# Image Registration of ^18^F-FDG PET/CT Using the MotionFree Algorithm and CT Protocols through Phantom Study and Clinical Evaluation

**DOI:** 10.3390/healthcare9060669

**Published:** 2021-06-04

**Authors:** Deok-Hwan Kim, Eun-Hye Yoo, Ui-Seong Hong, Jun-Hyeok Kim, Young-Heon Ko, Seung-Cheol Moon, Miju Cheon, Jang Yoo

**Affiliations:** 1Department of Nuclear Medicine, Veterans Health Service Medical Center, Seoul 05368, Korea; kimzzang978@naver.com (D.-H.K.); favoryu86@gmail.com (E.-H.Y.); euisung2637@gmail.com (U.-S.H.); akdlqmdl@bohun.or.kr (J.-H.K.); kossaem@hanmail.net (Y.-H.K.); diva1813@naver.com (M.C.); 2General Electronic Healthcare, Seoul 04637, Korea; seungcheol.moon@ge.com

**Keywords:** MotionFree algorithm, PET/CT misregistration, respiratory motion artifact, phantom test

## Abstract

We evaluated the benefits of the MotionFree algorithm through phantom and patient studies. The various sizes of phantom and vacuum vials were linked to RPM moving with or without MotionFree application. A total of 600 patients were divided into six groups by breathing protocols and CT scanning time. Breathing protocols were applied as follows: (a) patients who underwent scanning without any breathing instructions; (b) patients who were instructed to hold their breath after expiration during CT scan; and (c) patients who were instructed to breathe naturally. The length of PET/CT misregistration was measured and we defined the misregistration when it exceeded 10 mm. In the phantom tests, the images produced by the MotionFree algorithm were observed to have excellent agreement with static images. There were significant differences in PET/CT misregistration according to CT scanning time and each breathing protocol. When applying the type (c) protocol, decreasing the CT scanning time significantly reduced the frequency and length of misregistrations (*p* < 0.05). The MotionFree application is able to correct respiratory motion artifacts and to accurately quantify lesions. The shorter time of CT scan can reduce the frequency, and the natural breathing protocol also decreases the lengths of misregistrations.

## 1. Introduction

^18^F-fluorodeoxyglucose positron emission tomography (^18^F-FDG PET) is a useful imaging modality for diagnosing cancer, staging work-up, treatment planning, post-treatment evaluation, and recurrence determination in oncology [1,2]. However, the image resolution is not sufficient to detect anatomical location and often shows coarse spatial resolution with high noise levels. Computed tomography (CT) images are taken together with PET images in order to compensate for the attenuation correction and scattering of PET images and to provide anatomical information with adequate spatial resolution. Multimodality scanning, such as PET/CT, can obtain coregistered anatomical and functional images in a single study [3]. Combined image data from PET/CT have been considered as complementary information, which allows PET to demonstrate metabolism and CT to accurately localize abnormal lesions. Additionally, a combined scanner can help improve the quantitation of functional images derived from more accurate attenuation and partial-volume corrections. These advantages have important significance for functional parameters, including the standard uptake value (SUV) of a target lesion. The SUV is one of the representative parameters of PET/CT images and is considered to be an objective indicator of tumor metabolism [4,5,6].

The measurement of SUV, however, is vulnerable to potential biases and variabilities for a multitude of reasons, such as respiratory motion artifacts during combined PET/CT acquisition [7,8,9]. A patient’s respiration can cause the lesion to be overestimated and deformed, leading to the misregistration of the fusion PET/CT images [10,11]. In particular, due to the vertical motion of the diaphragm, the positional displacement and distortion of the lesions located at the base of the lungs and the upper segment of the liver are more likely to occur during the examination. In this situation, the SUV of a target lesion can decrease or the measurement of metabolic volume can increase compared to their actual values. To overcome this problem, various breathing instructions and methods have been investigated through many studies [12,13,14,15,16,17,18]. The respiratory motion artifacts can be reduced by gating PET images in correlation with respiration; the PET images are sorted into multiple time bins synchronized with the patient’s respiratory cycle or external devices [19,20,21,22]. For these methods of respiratory gating of PET acquisition, additional time would be required due to the prolonged scanning, and this approach has some limitations in actual clinical practice. 

A gating signal can be extracted directly from the acquired projection data using a variety of methods, commonly referred to as data-driven gating (DDG), which does not require an external gating device. DDG is known as an optimal software technique to detect respiratory motion within PET data, using the static phase for reconstruction [23,24]. 

While the DDG method can minimize respiratory motion artifacts, this technique does not take into consideration the breathing differences during CT image acquisition for attenuation correction. Since the differences in CT image acquisition according to a patient’s breathing can lead to PET/CT misregistration, it is also important to figure out which breathing protocol is most likely to cause the least PET/CT misregistration. Therefore, we demonstrated the degree of misregistration between PET images using the DDG software and CT images with regard to breathing methods, and evaluated which method is the most appropriate in clinical practice. We also aimed to investigate the benefits of the DDG algorithm through phantom tests. 

## 2. Materials and Methods

### 2.1. Phantom Experiments

Phantom experiments with a National Electrical Manufacturers Association/International Electrotechnical Commission (NEMA IEC) Body Phantom Set were performed using four spherical sources, with diameters of 10, 22, and 37 mm, and a vacuum single vial (20 mm), with an activity of 37 MBq of ^18^F-FDG. To simulate the cranio-caudal motion of abdominal organs during respiration, a real-time position management (RPM) moving phantom was driven forward and back, mimicking human breathing. The maximum displacement from the central position was set to ±10 or ±15 mm. Each of the sources pivoted at one end, while the other end oscillated in the Z-direction, along the PET gantry axis. Table 1 shows the PET/CT protocol for the phantom tests. 

The RPM system (Varian Medical Systems, Palo Alto, CA, USA) was used to track the position of a marker that was located on the respiratory motion platform. To evaluate the smearing effects as a function of motion, transaxial PET images were reconstructed along the rod source, and each phantom was filled and scanned in three separate phases as follows: static and with or without the DDG algorithm (MotionFree^TM^, GE Healthcare, Waukesha, WI, USA) for motion compensation. For each phantom experiment, the maximum SUV (SUVmax), mean SUV (SUVmean), and the volumes of the spheres were measured, and we investigated the distribution of activity using an AW workstation (Volume Viewer version 14.0, GE Healthcare). 

### 2.2. Patient Selection

This study included patients who had been referred for cancer staging work-ups (e.g., for lung cancer, colon cancer, head and neck cancer, or esophageal cancer), treatment response evaluations, or suspicions of disease recurrence using ^18^F-FDG PET/CT from September 2020 to December 2020. This prospective study was approved by the Institutional Review Board of VHS Medical Center (IRB No. 2020-07-017). All participants received and signed an informed patient consent form. 

The research was conducted by assigning patients to groups for three types of breathing protocol and two CT scanning times. One hundred patients were consecutively enrolled into each of the six groups, and a total of 600 patients participated in this study. Regardless of the patient’s age, sex, purpose of PET/CT examination, and type of cancer, patients were randomly selected and analyzed in order to investigate the PET/CT misregistration according to breathing protocols. 

### 2.3. Respiratory Gated PET Images and Breathing Protocols in CT Image Acquisition

After obtaining informed consent from patients, they were trained to breathe according to three protocols as follows: (a) patients who were not given any breathing instructions during image acquisition; (b) patients who were instructed to hold their breath after expiration during CT scan; and (c) patients who were instructed to continue natural breathing. The CT scan protocol was conducted at two different scanning times of 13 and 8 seconds (s) to determine how much misregistration of fusion PET/CT images can occur depending on the CT acquisition time. 

All patients were required to fast for at least 6 h to keep blood glucose below 200 mg/dL. Imaging acquisitions were performed approximately 60 min after the intravenous administration of 3.7 MBq/kg of ^18^F-FDG. The respiratory gated technique was applied for all participants using standard PET/CT (Discovery Molecular Imaging Digital Ready, GE Healthcare, USA) with the MotionFree algorithm based on principal component analysis, which could detect the respiratory waveform and utilize a Fourier transform to identify the strength of respiratory motion using an unsupervised machine learning technique [25,26,27]. 

CT images were obtained using 16-slice helical CT with the following settings: 120 keV, 50–110 or 50–80 mAs with Auto A mode, and a slice thickness of 3.75 mm. All other CT parameters including the dose length product (DLP) were set identically, and the scan time was set as 0.984:1 (13 s) or 1.531:1 (8 s) with different pitch variables. PET images were acquired from the head to the thigh, and attenuation-corrected PET images with 6–8 sequential bed positions (110 s per bed position, 192 × 192 matrix size) were reconstructed using a Q.Clear algorithm (Table 2). 

### 2.4. Image Analysis

The gated- or non-gated PET images of the phantom studies were investigated relative to the static images for the four phantoms. Since the spherical sources oscillated vertical to the axis of the gantry, each transaxial slice in the reconstructed image corresponded to a cross-section subject to a different vibrational amplitude. Considering the static image as a reference, we evaluated the differences between the volume and SUV values (e.g., SUVmax and SUVmean) of the phantom images when MotionFree was applied or not applied according to each sphere diameter and for the vacuum vials. 

The areas most affected by respiratory motion artifacts are observed in the lower lung field, liver dome, and upper abdomen during ^18^F-FDG PET/CT examination. On the other hand, the upper area of the lung is well known to be less subject to respiratory motion compared with the lower lung field. The most common type of respiratory artifact can cause curvilinear cold areas, which result in a downward displacement of the diaphragm due to lung expansion [28]. In the clinical study, the length of PET/CT misregistration was measured using the displacement in craniocaudal direction based on curvilinear cold areas in the coronal images (Figure 1). It was recorded in units of millimeters (mm), and we defined that a misregistration occurred when it exceeded 10 mm [29]. To determine whether the length of misregistration differs according to the type of breathing protocol and the CT scanning time, we compared the lengths in patients showing more than 10 mm of PET/CT misregistration. The tangent points were manually positioned to match the curvilinear shape of the liver border to the diaphragmatic dome. All measurements were performed by the same technologist to ensure reproducibility and avoid inter-observer variability. This procedure was performed on coronal images of fusion PET/CT using the Blend mode of an AW workstation (Volume Viewer version 14.0, GE Healthcare, USA) in order to visually identify the PET/CT misregistration well. 

### 2.5. Statistical Analysis

In the phantom study, a comparison of Pearson correlation coefficients was used to evaluate whether the difference between MotionFree and non-MotionFree data was statistically significant, using a static result as a reference value. All continuous variables of PET/CT misregistration were recorded as means ± standard deviations (SDs) or medians with interquartile ranges (IQRs) according to the Kolmogorov–Smirnov test. If this test showed a normal distribution, one-way ANOVA was performed. If continuous variables did not have a normal distribution, on the other hand, the Kruskal–Wallis test was applied to compare the quantitative features. Categorical variables were described as frequencies and analyzed using chi-square and comparison of proportions tests using the MedCalc software package (Ver. 9.5, MedCalc Software, Mariakerke, Belgium). A *p*-value less than 0.05 was considered statistically significant. 

## 3. Results

### 3.1. Phantom Study

Figure 2 shows the effect of the MotionFree algorithm on the activity distribution of spherical sources. For the NEMA IEC body phantom, the images produced by the MotionFree algorithm were in excellent visual agreement with the static images. The calculated results are detailed in Table 3.

In the cases in which the MotionFree algorithm was not applied, the radiotracer images were observed to have more blurring on visual assessment. The volumes also tended to be much larger, and the SUV parameters were much lower than those of the static images. After a comparison of Pearson correlation coefficients, the results of applying MotionFree were significantly closer to the static findings (*p* < 0.001), which suggested that improvements in both qualitative and quantitative image analyses were noticeable with the application of the MotionFree algorithm. 

### 3.2. Patient Characteristics

Six hundred patients were included in this prospective research study. Their baseline characteristics and PET/CT indications are listed in Table 4. A total of six groups were created according to the three types of breathing protocols and two CT scanning times, and 100 patients were randomly assigned to each group. The patients had sufficient cognitive abilities to understand the instructions about breathing methods by the PET/CT technologists, and they were blinded to their group assignments. 

### 3.3. PET/CT Misregistration

Table 5 lists the results of the patient study. In the clinical evaluation, 123 PET/CT misregistration cases out of 300 occurred in the 13 s scanning time, whereas 96 out of 300 were observed in the 8 s scanning time (41% vs. 32%, *p* < 0.05). The frequency of PET/CT misregistrations in the type (a) breathing protocol with the 13 s scanning time was significantly lower compared to other breathing protocols (*p* < 0.05 and *p* < 0.01, respectively). Although in the type (c) breathing protocol at 13 s, the frequency of misregistration was greatest at 51 cases, the median length of misregistration was the shortest at 16.6 mm. The misregistration lengths of type (c) were statistically shorter than those of types (a) and (b) (both *p* < 0.05).

At the 8 s scanning time, the frequency of PET/CT misregistration in the type (c) breathing protocol was the least with 18 cases, and this differed significantly from the other types (both *p* < 0.01). The median length was 12.6 mm, which only showed statistical significance from the type (b) breathing protocol after post-hoc analysis. Figure 3 illustrates the results of comparative analysis for the length of PET/CT misregistration shown in each breathing protocol according to the CT scanning time. 

When applying the type (c) breathing protocol, decreasing the CT scanning time significantly reduced the frequency and length of PET/CT misregistration (both *p* < 0.05). Even if the CT scanning time was reduced, there was no significant effect on the frequency of misregistration in types (a) and (b) (*p* = 0.29 and 0.89, respectively). 

## 4. Discussion

The aim of this research was to perform a clinical evaluation of the MotionFree algorithm, recently proposed by GE Healthcare, using phantom and patient studies. Our results demonstrated a strong association and excellent visual agreement between static and MotionFree-applied images in the NEMA IEC body phantom. We observed several important effects when the MotionFree algorithm was applied. First, it had the effect of reducing the overall shape of the activity distribution change due to respiratory movement. Second, it could show the effect of preventing the overestimation of source size. Finally, it could also prevent the drop in activity, allowing more accurate SUV measurements. 

Although several studies have been conducted on similar topics [30,31,32], our research has some strengths over those previous studies. One of the most valuable advantages is that we prospectively enrolled the largest number of patients to date, comprising 600 patients with various types of cancer. For this reason, we believe that our results can show more practical application in the actual clinical situations. Another advantage is that we conducted a comparative analysis with static images in a phantom study along with quantitative analysis and visual assessment. In addition to demonstrating PET/CT misregistration for three types of breathing protocols, we also found CT scanning time to be a factor that was highly influential in the current study. Our findings revealed that even a slight reduction in CT scanning time can minimize PET/CT misregistration in patients. 

Analysis of PET images acquired using the MotionFree algorithm revealed that these gating signals alleviated the respiratory artifacts and provided superior image quality as compared to images that were not corrected using MotionFree. Thus, the algorithm can be helpful for the characterization and quantitative assessment of real pathologic lesions of patients, providing benefits for image interpretation in clinical practice. Additionally, as we mentioned above, the total frequency of PET/CT misregistration could be reduced by using 8 s of scanning time rather than 13 s. Although there was no significantly different frequency in the type (a) and (b) breathing protocols depending on which CT scanning time was selected, in type (c), the frequency of misregistration could be greatly reduced when the 8 s scanning time was used. Natural breathing instruction by PET/CT technologists can provide consistent respiratory motion and lead to more regular movement for patients during CT scanning. 

One unresolved issue in our findings is that when the 13 s scanning time was adopted, the frequency of misregistration was significantly more in the type (c) breathing protocol compared to that of type (a). We presume that the definition for PET/CT misregistration may have affected this result. Since this is a single-center study, it is planned to acquire multi-center data to determine more general criteria for PET/CT misregistration classification. Therefore, it is necessary that the standard definition for misregistration should be evaluated through more research in the future. This study has also several limitations. First, it mainly included elderly patients. Thus, it is possible that the respiratory instruction may not have been properly performed. Another potential limitation is that there has been no investigation of each patient’s pulmonary history or the presence of current lung disease, such as pneumonia, which may affect the individual respiratory movements. Due to the unnecessary radiation exposure problem, there may be some limitations in accurate evaluation for our findings, as they were not obtained by taking repeated scans of the same patient under the various breathing protocols. Based on these limitations, there is a need for further study using more specific patient characteristics and a wider age range.

In conclusion, the MotionFree application was able to correct respiratory motion artifacts and to accurately quantify lesions in a phantom study. With regard to the patient study, the shorter time of CT scan (e.g., 8 s) can reduce the frequency of PET/CT misregistration, and natural breathing instruction by PET/CT technologists can also decrease the length of misregistration, which may affect clinical evaluation. 

## Figures and Tables

**Figure 1 healthcare-09-00669-f001:**
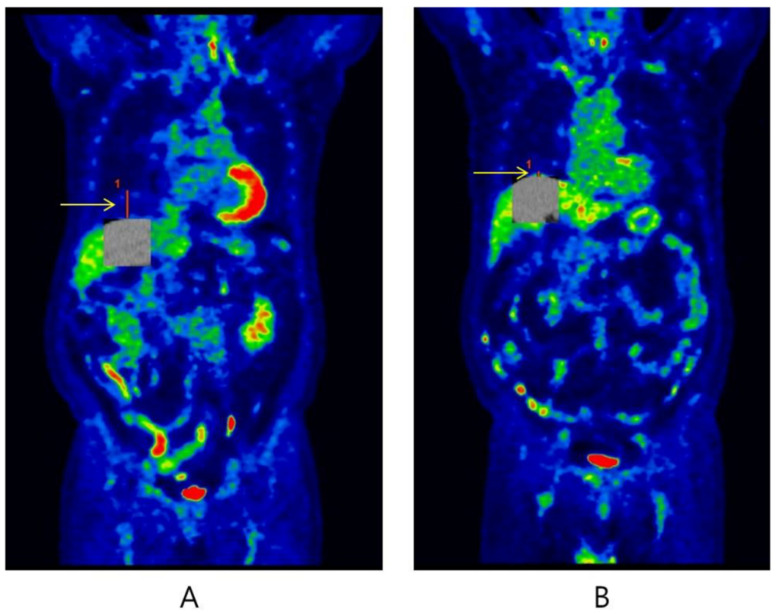
Coronal sections of ^18^F-FDG PET/CT images indicating the length of misregistration (red line) using the Blend mode of the AW workstation (Volume Viewer version 14.0, GE Healthcare, USA). Case (**A**) was 68 years old male patient with lung cancer, who performed type (a) breathing protocol at 13 s CT scanning time. The length of PET/CT misregistration was 39.0 mm. Case (**B**) was 78 years old male patient with lung cancer, who performed type (c) breathing protocol at 8 s scanning time. The length of misregistration was 7.3 mm.

**Figure 2 healthcare-09-00669-f002:**
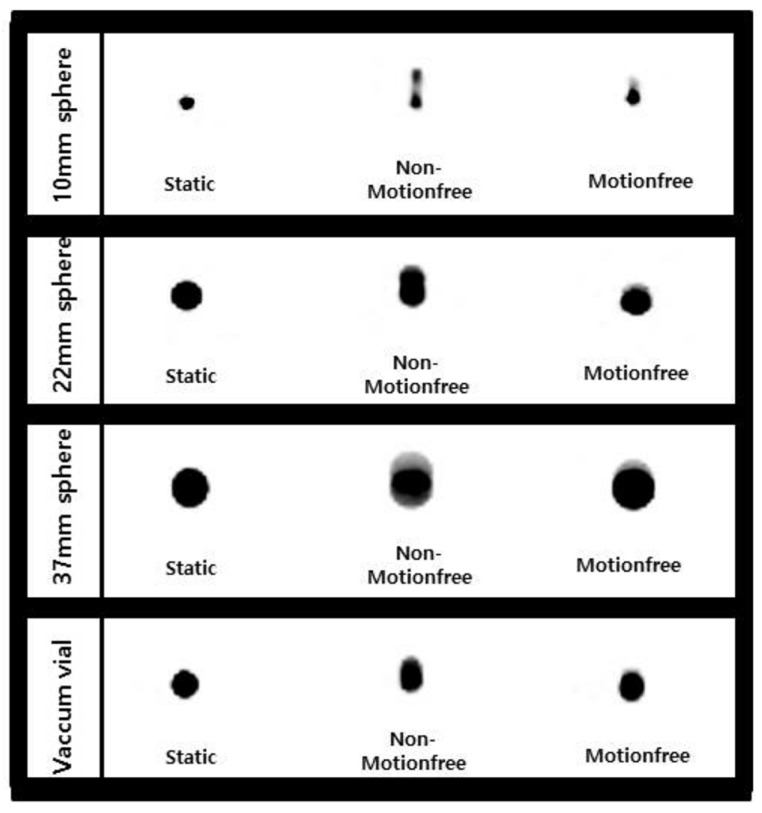
The effects of different distributions of activity according to the MotionFree application in a phantom study.

**Figure 3 healthcare-09-00669-f003:**
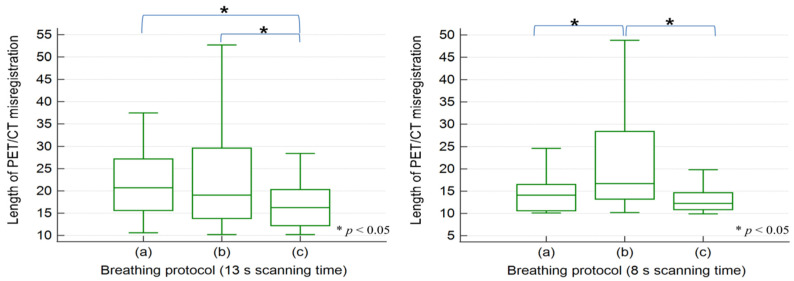
Statistical analysis of the lengths of PET/CT misregistrations for various breathing protocols and CT scanning times.

**Table 1 healthcare-09-00669-t001:** ^18^F-FDG PET/CT protocol for phantom test.

CT protocol	kVp	mA (Smart mA)	Rotation time	Pitch
	120	Min (50)~Max (80)	0.5 s	0.984:1
PET protocol	Base bed time	Q.Static bed time	Reconstruction	Matrix size
	2 min	4 min	Q.Clear 500	192 × 192

**Table 2 healthcare-09-00669-t002:** ^18^F-FDG PET/CT protocol for patient scanning.

CT protocol (13 s scanning time)	kVp	mA (Smart mA)	Rotation time	Pitch
	120	Min (50)~Max (80)	0.5 s	0.984:1
CT protocol (8 s scanning time)	kVp	mA (Smart mA)	Rotation time	Pitch
	120	Min (50)~Max (110)	0.5 s	1.531:1
PET protocol	Base bed time	Q.Static bed time	Reconstruction	Matrix size
	1 min 50 s	2 min 40 s	Q.Clear 500	192 × 192

**Table 3 healthcare-09-00669-t003:** The results of the phantom study.

Phantom	PET Parameter	Static	MotionFree (Pearson’s Correlation Coefficient: 0.9431)	Non-MotionFree (Pearson’s Correlation Coefficient: 0.6834)
10-mm sphere	Volume (cm^3^)	0.60	0.69	1.52
	SUVmax	31.93	22.19	11.74
	SUVmean	14.74	11.10	4.76
22-mm sphere	Volume (cm^3^)	5.87	7.48	10.34
	SUVmax	36.85	34.68	31.04
	SUVmean	26.93	20.56	15.03
37-mm sphere	Volume (cm^3^)	27.95	31.51	39.94
	SUVmax	20.49	18.74	18.36
	SUVmean	15.13	12.70	9.87
Vacuum vial	Volume (cm^3^)	11.69	12.78	16.95
	SUVmax	20.28	19.44	17.66
	SUVmean	13.76	12.19	9.03

SUVmax, the maximum standard uptake value; SUVmean, the mean standard uptake value.

**Table 4 healthcare-09-00669-t004:** Patient characteristics.

Characteristics	Types	Data
Age (years)		74 ± 7.3
Sex	Male	537 (89.5%)
	Female	63 (10.5%)
Body mass index		23.6 ± 3.3
PET/CT indication	H & N cancer	23 (3.8%)
	Lung cancer	290 (48.4%)
	GI cancer	117 (19.5%)
	Hepatobiliary cancer	75 (12.5%)
	Urinary cancer	11 (1.8%)
	Others	84 (14.0%)

H & N, head and neck; GI, gastrointestinal.

**Table 5 healthcare-09-00669-t005:** The frequency of PET/CT misregistrations and their lengths according to breathing protocol and CT scanning time.

CT Scanning Time	Results	Breathing Protocol		
		(a)	(b)	(c)
13 s	Frequency (*n*)	29 *^, †^	43 *	51 ^†,^ ^§^
	Length of misregistration (mm, median with IQR)	20.7 *^, ‡^ (15.6~27.2)	19.2 ^†^ (13.8~29.8)	16.6 *^, †, §^ (12.5~20.3)
8 s	Frequency (*n*)	36 *	42 ^†^	18 *^, †, §^
	Length of misregistration (mm, median with IQR)	14.3 *^, ‡^ (10.9~17.4)	17.0 *^, †^ (13.2~28.4)	12.6 ^†, §^ (11.1~15.8)

IQR, interquartile range; *, ^†^, ^‡^, ^§^ *p* < 0.05.

## Data Availability

Data available on requests from the corresponding author.

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
