# Peer review of "Image Registration of 18F-FDG PET/CT Using the MotionFree Algorithm and CT Protocols through Phantom Study and Clinical Evaluation"

_healthcare, 2021, doi:10.3390/healthcare9060669_

Round 1

Reviewer 1 Report

This paper investigates breathing motion in PET/CT and includes:

  • an experimental study with phantoms evaluating a commercial software for correction of respiratory motion during PET acquisition
  • a prospective study with 600 patients in which CT breathing protocol was controlled and misregistration between PET and CT was demonstrated

I really appreciate the effort of the authors and the difficulty to enrol such a huge amount of patients. However, I really do not see the relationship between the experimental and the clinical part. Of course, motion is a problem in PET/CT as patient can move either during PET acquisition or between PET and CT. In this paper, the experimental part evaluates a software to remove motion artifacts due to patient breathing during PET, but, as far as I understand, the clinical part evaluates movement between PET and CT. Two completely different problems.

If authors want to present all these results together in a single paper, I would suggest them to better justify the study design, to connect the experimental and clinical parts and to try to extract conclusions combining both. I would also suggest to change the title to reflect the actual content of the paper.

Minor comments:

  • References [1,2] should be updated to a more recent ones
  • The headings of Tables 1, 2 and 5 have been wrongly formatted
  • Replace SUVave by SUVmean, as this is the usual nomenclature
  • Phantom experiments with a National Electrical Manufacturers Association/International Electrotechnical Commission (NEMA IEC) Body Phantom Set were performed using 4 cylindrical sources, with diameters of 10, 22, and 37 mm, and a vacuum single vial (20 mm), with an activity concentration of 37 MBq of 18F-FDG”: Sources are spherical, not cylindrical, and the word “concentration” should be removed
  • In the results section, most data for misregistration area repeated in the text, the table and the graphs. If data is presented in tables and/or graphs that are cited in the text, there is no need to repeat the numbers in the text.
  • Are groups paired by age/gender?
  • In the clinical part, I really do not see the cause/effect between the breathing protocols during CT and the misregistration between PET and CT. Some patients move between PET and CT but this can happen randomly, and totally independently of what has happened during CT acquisition. Please, discuss.

Author Response

Dear editor and reviewers of Healthcare

We greatly appreiciate the review of our paper and the helpful suggestions. Please find below out point-by-point response to the reviewers’ and editor’s comments, and a decision of the changes made to the manuscript.

Sincerely,

Jang Yoo, M.D., Ph.D.

Departmet of Nuclear Medicine, Veterans Health Service Medical Center

Responses to Reviewer #1

This paper investigates breathing motion in PET/CT and includes:

  • an experimental study with phantoms evaluating a commercial software for correction of respiratory motion during PET acquisition
  • a prospective study with 600 patients in which CT breathing protocol was controlled and misregistration between PET and CT was demonstrated

I really appreciate the effort of the authors and the difficulty to enrol such a huge amount of patients. However, I really do not see the relationship between the experimental and the clinical part. Of course, motion is a problem in PET/CT as patient can move either during PET acquisition or between PET and CT. In this paper, the experimental part evaluates a software to remove motion artifacts due to patient breathing during PET, but, as far as I understand, the clinical part evaluates movement between PET and CT. Two completely different problems.

If authors want to present all these results together in a single paper, I would suggest them to better justify the study design, to connect the experimental and clinical parts and to try to extract conclusions combining both. I would also suggest to change the title to reflect the actual content of the paper.

  • We understand your concern. As you pointed out, the phantom study was conducted to demonstrate the usefulness of MotionFree algorithm and this proved that applying MotionFree algorithm in PET images can increase the accuracy of lesion evaluation. However, MotionFree algorithm can be only applied when acquring PET images. In actual clinical practice using fusion PET/CT imaging modality, PET/CT misregistration between the MotionFree applied PET and CT images may occur. Therefore, the authors of this study tried to find out whether PET/CT misregistration occurs less by controlling various breathing methods and CT imaging time.

As you said, since experimental and clinical parts are described on a single paper, it may seem that the correlation is low. I think the results of current research show that the quality of fusion PET/CT images can be improved by applying MotionFree algorithm when acquring PET and adjusting breathing and CT scan time when acquring CT images.

To better description the meaning of the overall content, I will change the title as follows: “Image registration of 18F-FDG PET/CT using MotionFree algorithm and CT protocols through phantom study and clinical evaluation”.

Minor comments:

  • References [1,2] should be updated to a more recent ones
  • As you suggested, we updated the references to more recent ones.

  • The headings of Tables 1, 2 and 5 have been wrongly formatted
  • As you pointed out, we have checked the wrong format. In my opinion, it is due to an error that occrred during the uploading process. We will modify and upload them in the correct format.
  • Replace SUVave by SUVmean, as this is the usual nomenclature
  • As you suggested, we have replaced SUVave by SUVmean.

  • “Phantom experiments with a National Electrical Manufacturers Association/International Electrotechnical Commission (NEMA IEC) Body Phantom Set were performed using 4 cylindrical sources, with diameters of 10, 22, and 37 mm, and a vacuum single vial (20 mm), with an activity concentration of 37 MBq of 18F-FDG”: Sources are spherical, not cylindrical, and the word “concentration” should be removed
  • As you suggested, we have replaced cylindrical by spherical and removed the word “concentration’.

  • In the results section, most data for misregistration area repeated in the text, the table and the graphs. If data is presented in tables and/or graphs that are cited in the text, there is no need to repeat the numbers in the text.
  • According to your valuable opinoin, we have reduced the unnecessary contents in the results section, so that more meaningful results can be described well.

  • Are groups paired by age/gender?
  • For this propective study, when we enrolled 100 patients from each group, we did not divide them according to age or gender. Due to the nature of our institute (Veterans hospital), elderly and male patients are relatively high. The composition of the average age and gender for each group are as follows.

(A) breathing protocol at 13 sec : 74.4 years old; male 89, female 11

(B) breathing protocol at 13 sec : 74.2 years old; male 92, female 8

(C) breathing protocol at 13 sec : 75.7 years old; male 90, female 10

(A) breathing protocol at 8 sec : 73.9 years old; male 90, female 10

(B) breathing protocol at 8 sec : 74.0 years old; male 87, female 13

(C) breathing protocol at 8 sec : 74.6 years old; male 89, female 11

This content was also described in the limitation section, and it is considered to be a point to be supplemented in future research.

  • In the clinical part, I really do not see the cause/effect between the breathing protocols during CT and the misregistration between PET and CT. Some patients move between PET and CT but this can happen randomly, and totally independently of what has happened during CT acquisition. Please, discuss.
  • We understand exactly what you are concerned. Of course, respiratory movements for CT scanning is completely independent from those acquring PET images. We assumed that PET images applying the MotionFree algorithm was corrected for the patient’s respiratory movement, and purposed to evaluate which methods can be performed to minimize PET/CT misregistration when acquring CT images. So, we aimed to adjust the CT scanning time and apply the breathing protocols to real patients.

Thank you for your helpful comments.

Reviewer 2 Report

In the paper "Clinical evaluation of image registration in 18F-FDG PET/CT using the MotionFree algorithm and breathing protocols", the authors evaluated the MotionFree algorithm effects in phantom and patient studies. 

The study with phantoms was useful to evaluate qualitatively and quantitatively the agreement of the images reconstructed with and without the algorithm "Motionfree" with the static ones using statistical analysis methods. The prospective patient study was used to evaluate the misregistration between PET images and CT images as a function of breathing protocols and CT scanning time. The results presented by the authors are interesting and provide useful information on what breathing protocols and CT scan times to use to minimize misregistration between PET and CT.

-In paragraph 2.1 Phantom experiments, 3rd line the authors write "4 cylindrical sources". Please change in "4 spherical sources".

-Figure 2 shows the effects of various activity distributions reconstructed with and without the "Motionfree" algorithm in spherical sources. 
The images of 22 mm sphere and 37 mm sphere show a blur effect from movement in the horizontal direction, while the images of 10 mm sphere and vacuum vial show a blur effect from movement in the vertical direction. The authors stated that the phantom movement was cranio-caudal or along the Z direction. What is the reason for this difference?

-Table 5, the last column is not well-formatted. The p-value in this column refers to which statistical test? 

-reference 5: please change "qunatification" in "quantification",

-reference 11: please change "acqusition" in "acquisition",

Author Response

Dear editor and reviewers of Healthcare

We greatly appreiciate the review of our paper and the helpful suggestions. Please find below out point-by-point response to the reviewers’ and editor’s comments, and a decision of the changes made to the manuscript.

Sincerely,

Jang Yoo, M.D., Ph.D.

Departmet of Nuclear Medicine, Veterans Health Service Medical Center

Responses to Reviewer #2

In the paper "Clinical evaluation of image registration in 18F-FDG PET/CT using the MotionFree algorithm and breathing protocols", the authors evaluated the MotionFree algorithm effects in phantom and patient studies.

The study with phantoms was useful to evaluate qualitatively and quantitatively the agreement of the images reconstructed with and without the algorithm "Motionfree" with the static ones using statistical analysis methods. The prospective patient study was used to evaluate the misregistration between PET images and CT images as a function of breathing protocols and CT scanning time. The results presented by the authors are interesting and provide useful information on what breathing protocols and CT scan times to use to minimize misregistration between PET and CT.

-In paragraph 2.1 Phantom experiments, 3rd line the authors write "4 cylindrical sources". Please change in "4 spherical sources".

à As you suggested, we have replaced cylindrical by spherical.

-Figure 2 shows the effects of various activity distributions reconstructed with and without the "Motionfree" algorithm in spherical sources.

The images of 22 mm sphere and 37 mm sphere show a blur effect from movement in the horizontal direction, while the images of 10 mm sphere and vacuum vial show a blur effect from movement in the vertical direction. The authors stated that the phantom movement was cranio-caudal or along the Z direction. What is the reason for this difference?

à As you pointed out, we have corrected the mistake in the Figure 2 and uploaded it again.

-Table 5, the last column is not well-formatted. The p-value in this column refers to which statistical test?  

à As you pointed out, we have checked the wrong format. In my opinion, it is due to an error that occrred during the uploading process. We will modify and upload it in the correct format. The p-value in the last column showed statistical significance among three groups after one-way ANOVA or Kruskal-Wallis test. However, this p-value may look a little confusion, and the post-hoc analysis result was also described as a footnote in the table, so it seems good to delete.

-reference 5: please change "qunatification" in "quantification",

à As you suggested, we have correted it.

-reference 11: please change "acqusition" in "acquisition",

à As you suggested, we have corrected it.

Thank you for your helpful comments.

Round 2

Reviewer 1 Report

Accept in present form

This manuscript is a resubmission of an earlier submission. The following is a list of the peer review reports and author responses from that submission.